# Spatial Distribution of Soil Heavy Metals and Associated Environmental Risks near Major Roads in Southern Tibet, China

**DOI:** 10.3390/ijerph19148380

**Published:** 2022-07-08

**Authors:** Wanjiang She, Linghui Guo, Jiangbo Gao, Chi Zhang, Shaohong Wu, Yuanmei Jiao, Gaoru Zhu

**Affiliations:** 1Faculty of Geography, Yunnan Normal University, Kunming 650000, China; swj143100808@163.com; 2School of Surveying and Land Information Engineering, Henan Polytechnic University, Jiaozuo 454000, China; guolinghui@hpu.edu.cn; 3Institute of Geographic Sciences and Natural Resources Research, CAS, Beijing 100101, China; gaojiangbo@igsnrr.ac.cn (J.G.); zhangchi@igsnrr.ac.cn (C.Z.); 4Laboratory of Transport Pollution Control and Monitoring Technology, Transport Planning and Research Institute, Ministry of Transport, Beijing 100028, China; zhugr@tpri.org.cn

**Keywords:** soil heavy metal pollution, ecological risk evaluation, health risk evaluation, southern Tibet, road engineering

## Abstract

Soil heavy metal pollution is becoming an increasingly serious environmental problem. This study was performed to investigate the contents of surface soil heavy metals (Cu, Zn, Pb, Cd) near six roads in the southern part of the Tibetan Plateau. Multivariate statistics, geoaccumulation index, potential ecological risk, and a human health assessment model were used to study the spatial pollution pattern and identify the main pollutants and regions of concern. The mean $I_{geo}$ was ranked in the order Cd > Cu > Zn > Pb, with the average concentrations of Cd, Zn, and Cu exceeding their corresponding background levels 4.36-, 1.00-, and 1.8-fold, respectively. Soil Cd level was classified as posing a considerable potential risk near national highways and a high potential risk near non-national highways, whereas soil Cu, Zn, and Pb were associated with a low potential ecological risk for each class of roads. Furthermore, the non-carcinogenic risk due to soil heavy metals for each class of roads was within the acceptable risk level for three exposure pathways for both adults and children, but the carcinogenic risk attributable to soil Pb exceeded the threshold for children near highways G318, G562, and G219 and for adults near highway G318. Our work not only underscores the importance of assessing potential threats to ecological and human health due to soil heavy metal pollution on road surfaces but also provides quantitative guidance for remediation actions.

## 1. Introduction

Soil heavy metals cannot be degraded readily by soil microorganisms and endanger ecosystems’ safety and human health through biomagnification [1]. With the rapid social and economic development, soil heavy metal pollution is becoming a serious environmental problem [2]. It has been reported that about 600,000 hectares of land in the USA are contaminated with heavy metals, and there are about 400,000 contaminated sites in Germany, the UK, Italy, Spain, and other European countries [3]. China’s soil heavy metal pollution level exceeds the standard rate of 16.1% [4], with Pb pollution being particularly prominent in the Yunnan–Guizhou area, the Pearl River Delta, and southeast Fujian [5,6], and As, Hg, Pb, and Cd contents being much higher than the national soil element background value in the southwestern region [7]. Therefore, clarification of the spatiotemporal variation in regional soil heavy metal pollution and its environmental hazards is a major scientific issue of general concern for the government and the scientific community.

Soil heavy metal pollution is closely related to human activities [8,9]. Automobile exhaust emissions, automobile tires, brake pad wear, and the burning of lubricating oil are important sources of heavy metal pollution in the soil around roads [10,11,12]. The characteristics of soil heavy metal pollution from traffic sources also depend on regional atmospheric circulation and running water. Some studies have shown that heavy metal contents in soil decreased exponentially with distance from the road, with Zn tending to reach the background level at 30 m [10,11]. Other studies have also reported an initial increase in soil heavy metal contents followed by a decrease with distance from the road, with Pb, Cd, and Cu contents peaking in the range of 25–50 m from the road [13] and approaching the background level at 70–150 m [14]. Accurately investigating spatiotemporal variation in heavy metal pollution in roadside soil, especially in environmentally sensitive areas, may provide a scientific basis for preventing and controlling environmental pollution.

A number of methods are available for evaluating soil heavy metal pollution, including the comparison method, geochemical method, and statistical analysis [15,16,17,18]. Further, there has been extensive research based on the Nemerov comprehensive pollution index method [15,16,17,18]. Some studies used random site indicators to quantify the morphology and evolution of soil pollution [19]. The geoaccumulation index, potential ecological risk assessment, and human health risk model are alternative approaches for identifying pollution levels and environmental hazards [8,20,21]. For example, the geoaccumulation index method can better reveal the degree of enrichment of exogenous heavy metal elements [22], and the potential ecological risk assessment method not only considers the characteristics of soil heavy metal content but also accounts for the biotoxicological characteristics of heavy metals, with the combination of both aspects allowing the effective representation of soil single heavy metal pollution and the integrated ecological hazards of heavy metals in soil [18,20,23,24] and providing strong support for ecological environment pollution control [24,25]. The human health risk assessment model assesses non-carcinogenic and carcinogenic risks for adults and children in different regions through the oral, inhalation, and dermal exposure routes [11,26].

As the “third pole of the earth,” the Qinghai–Tibet Plateau is the origin of the Yellow River, Yangtze River, and Lancang River and exerts an important influence on the ecological environmental security of China. The ecological environment of the Qinghai–Tibet Plateau is fragile and unique, can thus be easily damaged, and is difficult to restore to its original state. Ecological security issues have become increasingly prominent in recent years with the rapid development of industrial and mining enterprises and increases in traffic levels [27]. Some studies have shown that most areas of the Qinghai–Tibet Plateau have moderate or high soil levels of Cd pollution [28,29,30]. The degree of pollution is greater in Qinghai than in Tibet [23]. Studies of soil heavy metals from transportation sources on the Qinghai–Tibet Plateau are mostly concentrated near some parts of highways G214, G109 [31], the Lin-la Highway [32], and roads around some lakes [30,33]. The assessment of the potential ecological and human health risks is still limited, with little information available regarding the possible impacts of road construction on the ecological environment.

In this study, we focused on major national highway and non-national highway projects in the southern part of the Qinghai–Tibet Plateau. The objectives of the study were to evaluate the concentrations of trace metals (Cu, Zn, Pb, and Cd) in selected locations and characterize the spatial variation in heavy metals in roadside soil, investigate pollution levels using the geoaccumulation index ($I_{geo}$) in regions of concern, and identify the comprehensive environmental risk from road construction projects using both potential ecological risk assessment and human health risk assessment. This work will improve the understanding of the environmental influence of road construction projects and provide important support for ecologically safe construction on the Qinghai–Tibet Plateau.

## 2. Materials and Methods

### 2.1. Study Area

The study area is located in the southern Tibetan region of the Qinghai–Tibet Plateau (26°50′–36°53′ N, 78°25′–99°06′ E) in China (Figure 1). The Qinghai–Tibet Plateau consists of precipitous terrain with interconnected mountains and valleys and hosts a fragile and unique ecosystem that can be easily disrupted and for which post-disturbance recovery to the original state would be difficult. Southern Tibet is an important gateway to China’s southwestern border; because it is rich in resources and undergoing rapid socioeconomic development, it serves as the main site of socioeconomic development in the Tibetan Plateau region. Highways G318, G219, G562, and G216 are part of a modern transportation network in southern Tibet. These highways promote the development of the regional economy and society and have greatly improved people’s quality of life. However, environmental pollution from road traffic has gradually begun to affect the lives of residents in the area [27].

### 2.2. Field Sampling and Sample Preparation

Based on satellite images, 14 sampling sites were selected in flat and open areas near important roads in southern Tibet and are shown in Figure 1. For each sampling site, we randomly set collection points 2, 5, 10, 15, 30, and 50 m from the road, and a total of 84 soil samples (about 500 g each sample) of surface soil (0~10 cm) were collected. All samples were naturally air-dried in the lab for more than 2 weeks and sieved through a 100-mesh nylon sieve (diameter about 0.15 mm) to remove stones, plant residues, and other debris. Then, each soil sample was uniformly mixed and extracted using the four-point method for laboratory determination. The soil Cu and Zn contents were determined using flame atomic absorption spectrophotometry (HJ491-2019) [34], after approximately 0.2~0.3 g of sample was digested in a digestion tank using an HCl/HNO_3_/HF solution in a microwave digestion furnace according to the digestion method of HJ 832 [35]. For soil Pb and Cd, approximately 0.1~0.3 g of sample was digested in a Teflon crucible using an HCl/HNO_3_/HF/HClO_4_ solution on an electric hot plate. Subsequently, heavy metal contents were determined using graphite furnace atomic absorption spectrophotometry (GB/T17141-1997) [36]. Each sample was measured by using an atomic absorption spectrophotometer (TAS-990AFC, Beijingpuxi), and the analysis results were reliable when the repeat sample analysis error was below 5% for the 10% of parallel samples and the two control samples for each type of heavy metal element were within effective range.

### 2.3. Geoaccumulation Index

The geoaccumulation index, $I_{geo}$, first introduced by Muller in 1969, represents the degree of enrichment of an exogenous heavy metal element in a study area affected by human activities [22]. The $I_{geo}$ has been widely used in trace studies [18,21,29,37] and is calculated as follows:(1)$$I_{geo}=\log_{2}\;[C_{i}/(kB_{i})]$$
where $C_{i}$ is the concentration of the heavy metal of interest in the soil (mg/kg), k is a coefficient representing fluctuations in the background level due to differences in the parent material in various areas (in general, k = 1.5), and $B_{i}$ (mg/kg) is the background concentration of the heavy metal. To avoid the assessed results having lower differentiation, the seven-grade classification method was selected for assessing the contamination levels of heavy metals in soils [18]. Based on a previous study by Men et al. [38], the pollution level was divided into the following seven grades: $I_{geo}$ ≤ 0, no pollution (Class 0); 0 < $I_{geo}$ ≤ 1, mild to moderate pollution (Class 1); 1 < $I_{geo}$ ≤ 2, moderate pollution (Class 2); 2 < $I_{geo}$ ≤ 3, moderate to heavy pollution (Class 3); 3 < $I_{geo}$ ≤ 4, heavy pollution (Class 4); 4 < $I_{geo}$ ≤ 5, heavy to extreme pollution (Class 5); 5 < $I_{geo}$, extreme pollution (Class 6). In this study, the background levels of elements in Tibetan soil were used as standards [39].

### 2.4. Ecological Risk Model

The Swedish scientist Hakanson proposed the potential ecological hazard index method [25], one of the international methods for the study of heavy metals in soils (sediments), to evaluate heavy metals in soils or sediments from a sedimentological point of view, based on the nature of the heavy metals and their behavior in the environment, such as transport, transformation, and deposition. In recent years, the potential ecological risk index (RI) has been applied in more and more research about the contamination assessment of multiple elements [38,40,41,42]. The RI was evaluated based on heavy metal toxicities and levels of heavy metals in the soil of southern Tibet to support local ecological management [25]. The relevant formulas are as follows:(2)$$C_{r}^{i}=C_{s}^{i}/C_{n}^{i}$$
(3)$$E_{r}^{i}=T_{r}^{i}C_{r}^{i}$$
(4)$$\mathrm{RI}=\sum_{i=1}^{n}E_{r}^{i}=\sum_{i=1}^{n}T_{r}^{i}C_{r}^{i}=\sum_{i=1}^{n}T_{r}^{i}C_{s}^{i}/C_{n}^{i}$$
where RI is the composite potential ecological risk index for the area, $E_{r}^{i}$ is the ecological risk index, $T_{r}^{i}$ is the toxicity coefficient of a given heavy metal, $C_{r}^{i}$ is the contamination factor of the heavy metal, and $C_{s}^{i}$ and $C_{n}^{i}$ correspond to the measured concentration and background level of heavy metal *i*, respectively. The $T_{r}^{i}$ values for Cu, Zn, Pb, and Cd are 5, 1, 5, and 30, respectively [43]. The magnitude of RI is related to the type and quantity of the evaluated pollutant; the higher the quantity of the pollutant, the stronger the toxicity, and the larger the RI value. Therefore, when applying the RI for ecological risk evaluation, it should be adjusted according to the type and quantity of the evaluated pollutants. In this paper, the calculation was based on the method of Ma et al. [44], and the adjusted $E_{r}^{i}$ and RI were categorized into various classes, as shown in Table 1.

### 2.5. Human Health Risk Model

The health risk assessment method recommended by the US Environmental Protection Agency (USEPA) was used to calculate the average daily exposure (ADD; mg·kg^−1^·d^−1^) via three exposure routes: oral ingestion, dermal contact, and inhalation [45]. In our study, the health risks associated with four heavy metals—Cu, Zn, Pb, and Cd—were assessed, and the relevant equations are as follows:(5)$$ADD_{ing}=\frac{C\times IR_{1}\times CF\times EF\times ED}{BW\times AT}$$
(6)$$ADD_{derm}=\frac{C\times CF\times SA\times AF\times ABS\times EF\times ED}{BW\times AT}\;$$
(7)$$ADD_{inh}=\frac{C\times IR_{2}\times EF\times ED}{BW\times PEF\times AT}$$
(8)$$HQ_{i}=\frac{ADD_{ij}}{RFD_{ij}}$$
(9)$$HI=\sum HQ_{i}$$
(10)$$CR=\sum ADD_{ij}\times SF_{ij}$$
where *C* in Equations (5)–(7) indicates the concentration of a given heavy metal in the soil. The values and physical significance of other relevant parameters are shown in Table 2 [46].

$HQ_{i}$ in Equations (8)–(10) is the non-carcinogenic risk resulting from a certain exposure route. *HQ* and the hazard index (*HI*) were applied to estimate the non-carcinogenic risk. *HI* < 1 indicates that adverse health effects are unlikely, whereas *HI* values exceeding 1 suggest potential non-carcinogenic health effects, and higher *HQ* values indicate a higher degree of human health risk [47]. The *HI* is the overall non-carcinogenic risk index, $RFD_{ij}$ is the maximum intake according to body weight per unit time that does not cause adverse reactions in the human body, the *CR* index is a measure of carcinogenic risk, $SF_{ij}$ is the slope coefficient of a given carcinogenic heavy metal for a particular different exposure route, and the reference values for $RFD_{ij}$ and $SF_{ij}$ are shown in Table 3 [46,48,49]. According to the USEPA, CR < 10^−6^ indicates no cancer risk, 10^−6^ < *CR* < 10^−4^ indicates that the risk is within the acceptable range, and *CR* > 10^−4^ indicates that the human tolerance level has been exceeded [38,50].

## 3. Results

### 3.1. Heavy Metal Distributions

Descriptive statistics for soil heavy metal concentrations near roads are listed in Table 4 and Table 5. As shown in Table 4, the average soil concentrations of soil Cd, Cu, and Zn on the roadsides of national highways (G318, G562, G219, G216) were 3.99-, 1.75-, and 1.00-fold the corresponding background values in southern Tibet soil, respectively. The values on the roadsides of non-national highways (Laliu Road and Gangpai Road) were 5.72-, 2.39-, and 0.99-fold the background values, respectively. In addition, spatial variation in the soil heavy metal content was ranked as Cd > Cu > Pb > Zn, with the degree of variation in soil Pb and Cu higher near national highways than near non-national highways. For example, in Table 5, the coefficients of variation (CVs) of soil Cu concentration near G318 and G562 were >40%, whereas they were only 6% and 9% near G216 and Laliu Road, respectively. By contrast, the CVs of soil Cd near G318, G562, G219, and Gangpai Road were >35%, roughly three times higher than at the other roadsides. These data exhibited a high level of variability, indicating that the levels of the three heavy metals are influenced by anthropogenic factors.

The spatial variation of soil Cu, Zn, Pb, and Cd differed significantly among different roads (Figure 2). Generally, soil Cu and Zn contents peaked within 10–20 m of national highways, whereas most of the peak values appeared at 50 m from non-national highways. The soil Pb concentration peaked within 2–30 m from both national and non-national highways, whereas the soil Cd content fluctuated markedly with distance from the road among different roads, with the peak value occurring within 5–50 m from the national highways and mostly at 15 and 50 m from the non-national highways.

### 3.2. Degree of Heavy Metal Pollution

According to the geoaccumulation index, the degree of soil heavy metal pollution was ranked as Cd > Cu > Zn > Pb (Figure 3). Soil Cu levels by both national and non-national highway roadsides were determined as Class 1 pollution levels. Soil Cd concentrations were classified as Class 2 (moderately polluted), whereas soil Zn and Pb pollution levels were generally lower than those of the other soil heavy metals and were classified as Class 0 (Figure 3b). Specifically, the degree of soil Cu pollution was Class 1 on the roadside of all roads except G219, whereas soil Cd pollution was relatively moderate on the roadside of the four national highways and moderate to heavy on the roadside of Gangpai Road. In comparison, regarding Zn and Pb, the roadside of all roads were generally considered unpolluted (Figure 3a).

For further quantitative assessment, the degree of soil pollution for each of the metals examined was analyzed at different distances from each road. As shown in Figure 4, soil Cu pollution was mild to moderate on the roadsides of highways G562 (except at 30 m), G318, G216, while the roadside of highway G219 was unpolluted and was classified as class 0. In addition, the roadsides of non-national highways were classified as mild to moderate polluted. Soil Cd was generally present at moderately polluted levels (Class 2), but the pollution level was categorized as Class 1 at 2 m from highway G318 and class 3 at 50 m from highway G219 and at 2, 15, and 50 m from Gangpai Road. Regarding soil Zn and Pb, sites at different distances from the national and non-national highways were classified as non-polluted, except for the roadside at 50 m from highway G219, which was classified as unpolluted to moderately polluted with respect to soil Zn (class 1).

### 3.3. Ecological Risk Assessment

Single-factor potential ecological risks due to individual soil heavy metals occurring on the roadside of two types of highways in southern Tibet are shown in Figure 5. In general, the mean $E_{r}^{i}$ values of soil Cu, Zn, and Pb on the roadside of the national and non-national highways indicated low potential ecological risk, but the $E_{r}^{i}$ values of soil Cd indicated high potential risk on the roadside of national highways and non-national highways (except for G318, G562), suggesting a high risk to the ecological environment. In general, soil Cd was at a considerable potential risk level on the roadside of national highways and at a high potential risk level on non-national highways’ roadsides (Figure 5a). Figure 5b shows that the single-factor potential ecological risks of soil Cu, Zn, and Pb on national and non-national highways were low potential, while soil Cd at 2 m, 10 m, and 50 m on the roadside from national highways showed a considerable potential risk, and at 5 m, 15 m, 30 m, a high ecological potential risk. A high potential risk level was shown at all distances on the side of the non-national highways.

In addition, as shown in Figure 6, the $E_{r}^{i}$ values of soil Cu, Zn, and Pb at different distances from the road still indicated a low potential ecological risk on the roadside of national and non-national highways. Whereas soil Cd was considered moderately ecologically hazardous at 2–15 m and at 50 m from the national highway G318, at 2 m, 10 m, 30 m–50 m from G562, and at 50 m from G216, a high potential risk was determined for the other national highways at different distances. We measured a high potential risk on the roadside of the non-national highway Laliu Road (except at 50 m), and a high potential risk at most distances from Gangpai Road, which appeared to be the road associated with the most serious potential ecological hazard.

As shown in Figure 7, the comprehensive potential ecological risk on the roadside of the national highways and non-national highways was considerable (Figure 7a). For each road, the comprehensive potential ecological risk of each soil heavy metal was classified as considerable (except at 15 m and 50 m from Gangpai Road, which was high) at different distances from each non-national highway. In comparison, the comprehensive potential ecological risk due to soil heavy metals at 50 m from G219 was high, whereas considerable potential ecological risk levels were determined for other areas on the roadsides of the national highways (Figure 7b).

### 3.4. Human Health Risk Assessment

As shown in Figure 8, for both adults and children, the HI related to the non-carcinogenic risk from exposure to soil Cu, Zn, Pb, and Cd through three exposure routes on the roadside of national and non-national highways was <1, and the non-carcinogenic risk associated with respiratory inhalation was much lower than that associated with oral ingestion and dermal absorption.

As shown in Figure 9, the *CR* values for exposure to soil Pb and Cd on the roadside of national and non-national highways through the dermal and respiratory routes were below the carcinogenic risk threshold of 10^−6^, but both adults and children were found to face a carcinogenic risk from oral exposure. For adults, the *CR* value for soil Pb on the roadside of G318 was 1.21 × 10^−4^, which exceeded the upper end of the carcinogenic risk threshold range by 1.21-fold, whereas the *CR* values for the soil on the roadside of G318, G562, and G219 for children were 2.82 × 10^−4^, 2.33 × 10^−4^, and 1.91 × 10^−4^, which exceeded the upper end of the carcinogenic risk threshold range by 2.82-, 2.33-, and 1.91-fold, respectively. The *CR* values for roadside soil Cd were within the carcinogenic risk threshold range for both adults and children, but children are more susceptible to heavy metal contamination compared to adults.

## 4. Discussion

The mean contents of three soil heavy metals, Cu, Zn, and Cd, found near roadsides in southern Tibet mostly exceeded the Tibetan background values; this was observed especially for Cd. Soil Cu on the roadside is mainly derived from the wear and tear of the vehicle’s brake plates and pads [51]. Soil Zn derives from tire lubricants and the corrosion of galvanized nickel automotive parts [52,53], and Cd comes from vehicle tires and fuel combustion [54,55]. The development of transportation has led to an increasing number of people traveling into Tibet, which results in more wear and tear than normal uniform driving and also increases the release of heavy metals [56]. Appendix A shows that there usually are many vehicles in real time on the road. Although the number of trucks differs from the number of other types of vehicles, the emissions from heavy vehicles are 5–6-fold those from light vehicles [57]. All together, these emissions undoubtedly lead to an increase in the soil heavy metal contents around roads [58,59]. 

Taking the distance from the road into account, the soil Pb (on the roadside of G562) and Cd (on the roadside of G216) contents exhibited a decreasing trend with increasing distance from the road, which is generally consistent with the results of Gao and Wang et al. [31,33] with regard to soil heavy metals on other roadside areas of the Qinghai–Tibet Plateau. The closer to the road, the greater the influence of the above factors on soil heavy metal content, which in turn leads to higher soil heavy metal contents closer to roads [58,59]. However, the soil Cu content increased with distance from G216 and Laliu Road, and Zn level also increased with distance from Laliu Road (Figure 2), which might be related to the abundance of clay minerals, carbonates, organic matter, and hydrated oxides as well as to certain physical conditions and traffic-related factors [60]. Heavy metal-containing particles emitted from road traffic can diffuse to more distant areas under the influence of the geographic location, climate, meteorology, and other conditions depending on the road location [61]. On the other national highways and non-national highways, the contents of soil Cu, Zn, Pb, and Cd were measured at a certain distance from the road and showed a skewed distribution, similar to the results of Ma et al. regarding soil heavy metals near Lianhuo Expressway [13]. Transportation processes and concentrations of heavy metals differ greatly under different environmental conditions, which should be further studied in the future.

A single-factor potential ecological risk coefficient ($E_{r}^{i}$) > 30 and a potential ecological risk index (RI) > 50 indicate potential risk to the local ecological environment from soil heavy metals [21,38]. In this study, three heavy metals, i.e., soil Cu, Zn, and Cd, near major roads in southern Tibet presented low potential ecological risk, whereas Cd was associated with considerable ecological risk near four national highways and high potential ecological risk near two non-national highways. Compared to the other three soil heavy metals, soil Cd contributed the most to the RI [16], largely due to its high toxicity [43]. The RI is usually controlled by heavy metals with high contents in soil or significant toxicity [18]. The ecological risk associated with soil Cd was also shown to be higher than that associated with other heavy metals in a study of Beijing road soils by Yu et al. [62].

This study showed that the non-carcinogenic risk from heavy metals entering the human body through the three exposure routes was within the acceptable range near major roads in southern Tibet. However, the carcinogenic risk was still not negligible; especially, the risk associated with soil Pb exposure through oral ingestion for children is much higher than for adults, and this risk was more obvious near the national highways. Soil Pb from traffic emissions had the greatest impact on the carcinogenic and non-carcinogenic risks [8]. Although the soil Pb content in the atmosphere has been reduced since the introduction of unleaded gasoline [63], the low mobility of Pb and its high affinity for organic matter in the soil make it easier for Pb to accumulate in soil [64]. Therefore, it is necessary to raise concern about the impact of soil Pb on human health. Children are more vulnerable to heavy metal pollution than adults [65,66]. Accordingly, health risks, including non-carcinogenic and carcinogenic risks, are generally higher for children than for adults [67]. Although both carcinogenic and non-carcinogenic risk levels from soil Cd exposure were within the acceptable range in this study, Cd has high biological accessibility, toxicity, and carcinogenicity and cannot be ignored in human health risk assessments for the development of preventative strategies [68].

Overall, there remain some uncertainties in this study. Firstly, in order to better determine the potential impact scope and extent of heavy metal pollution from road construction projects in the southern part of the Tibetan Plateau, we only focused on flat and open areas with about 100 m width on roadsides as sampling sites. A complex terrain environment made it difficult to find enough ideal sites, and the total of 14 sampling sites may cause some limitations in the final assessment. On the other hand, future work could focus on position monitoring of soil heavy metals to improve our understanding of soil heavy metals’ migration mechanisms along roadsides.

## 5. Conclusions

Heavy metal concentrations at selected locations around southern Tibet were determined. Soil Cu, Zn, and Cd contents near both national and non-national highways exceeded the background values, particularly near Gangpai Road, a non-national highway. There were also noticeable differences in the distributions of four heavy metals—Cd, Cu, Zn, and Pb—with distance from the road between national and non-national highways. Based on the geoaccumulation index, the pollution level due to the four heavy metals was ranked in the order of Cd > Cu > Zn > Pb. There was no pollution attributable to soil Zn and Pb, whereas Cu and Cd pollution was greater near non-national highways than near national highways, with Cd pollution being especially high near Gangpai Road. The levels of soil Cu and Cd pollution at different distances from the road were consistent with the overall evaluation results, according to which Cd pollution at 15 and 50 m from the non-national highways was more serious than at the same distances from the national highways. The single-factor potential ecological risk ($E_{r}^{i}$) from soil Cu, Zn, and Pb was generally low, the potential risk from Cd near national highways was considerable, and the potential risk near non-national highways was high, especially near Gangpai Road. In addition, the potential ecological risk was higher near non-national highways than near national highways. The overall results of the ecological risk evaluation for the four heavy metals at various distances from the road were generally consistent. In addition, the mean combined potential ecological risk (RI) level for both national highways and non-national highways was considerable. Based on a human health risk assessment model, the non-carcinogenic risk from exposure to heavy metals in the soil near southern Tibet’s main highways via three exposure routes was within the acceptable range for adults and children, but the carcinogenic risk was not negligible. The carcinogenic risk from soil Pb entering the human body via the oral route was much higher for children than for adults and was more severe near national highways. It is recommended that the relevant authorities raise awareness on the impacts of soil Cd pollution in the region.

## Figures and Tables

**Figure 1 ijerph-19-08380-f001:**
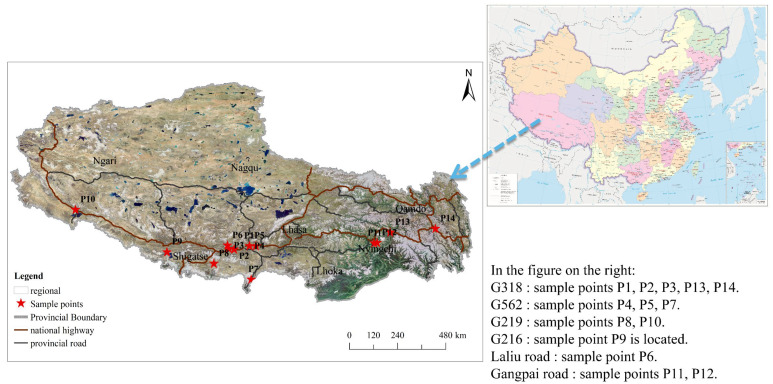
Locations of the sampling points in the study area.

**Figure 2 ijerph-19-08380-f002:**
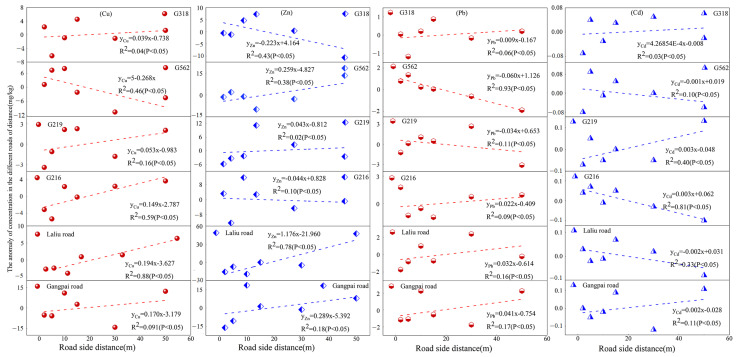
Soil heavy metal contents in relation to distance from the road.

**Figure 3 ijerph-19-08380-f003:**
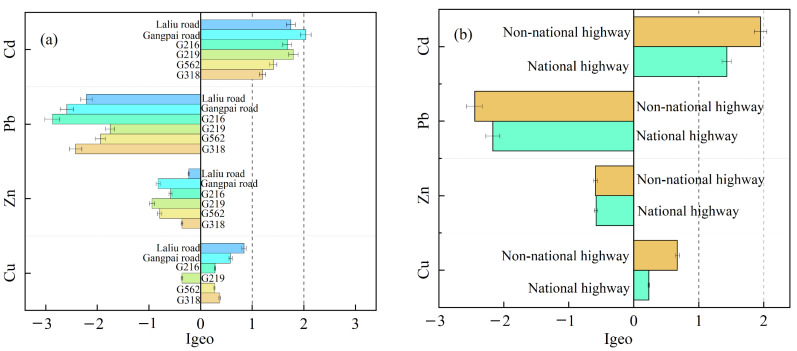
Mean geoaccumulation index for soil heavy metals. (**a**) The mean $I_{geo}$ for soil heavy metals near different roads. (**b**) The mean $I_{geo}$ for soil heavy metals near national and non-national highways.

**Figure 4 ijerph-19-08380-f004:**
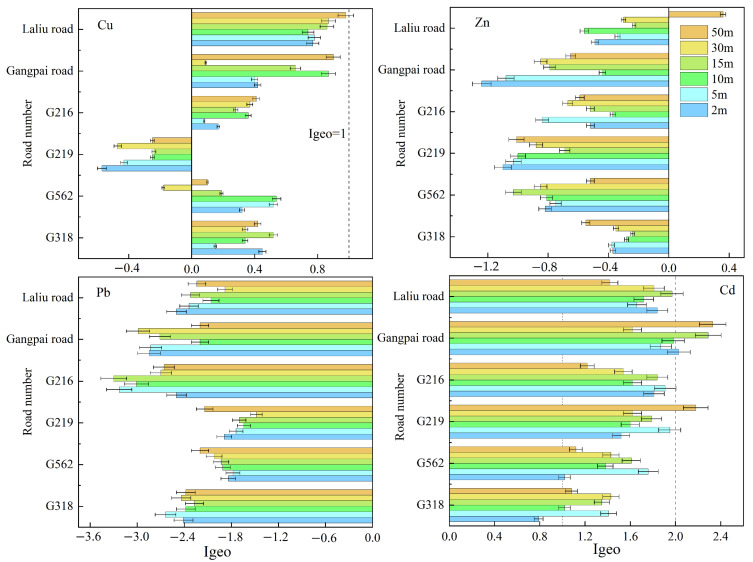
Geoaccumulation index of soil heavy metals at different distances from each road.

**Figure 5 ijerph-19-08380-f005:**
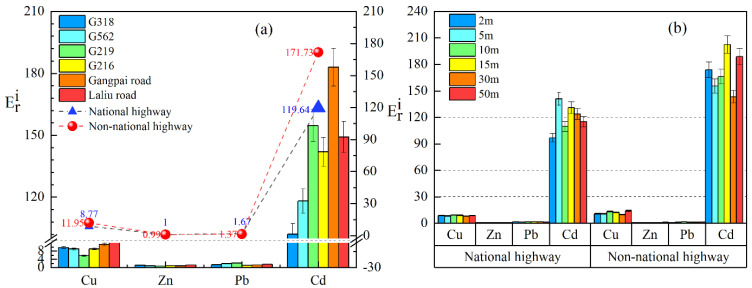
Single-factor potential ecological risk coefficients ($E_{r}^{i}$). (**a**) The average $E_{r}^{i}$ values of soil heavy metals in different roads. (**b**) The average $E_{r}^{i}$ values of soil heavy metals at different distances from national and non-national highways.

**Figure 6 ijerph-19-08380-f006:**
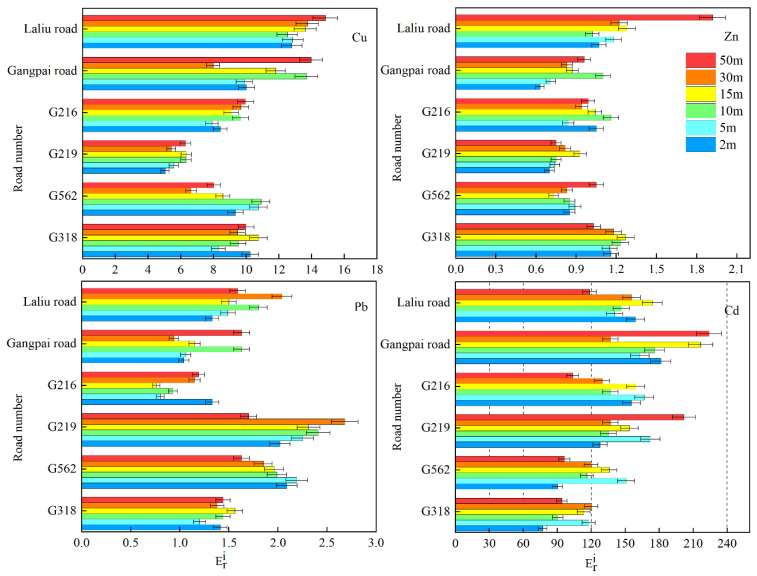
Average $E_{r}^{i}$ values representing soil heavy metal pollution at different distances from each road.

**Figure 7 ijerph-19-08380-f007:**
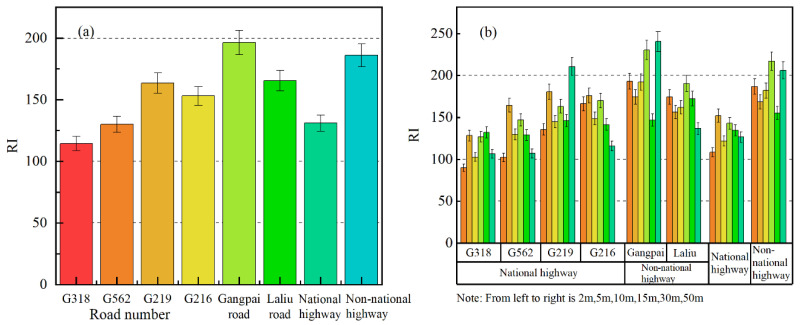
Potential ecological hazard index (RI). (**a**) RI for each road (national and non-national highways). (**b**) RI at different distances from each road.

**Figure 8 ijerph-19-08380-f008:**
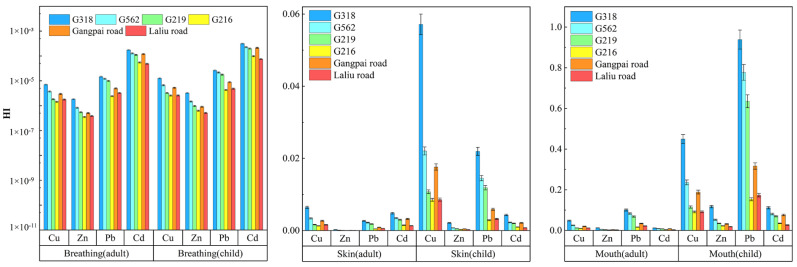
Non-carcinogenic risk from soil heavy metals found near different roads for children and adults.

**Figure 9 ijerph-19-08380-f009:**
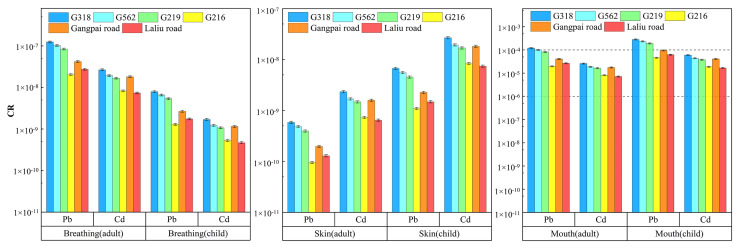
Carcinogenic risk from exposure to soil heavy metals near different roads for children and adults.

**Table 1 ijerph-19-08380-t001:** Classification of potential ecological risk.

$E_{r}^{i}$	RI	Class of Ecological Risk
$E_{r}^{i}$ < 30	RI < 50	Low potential ecological risk
$30\;\leq \;E_{r}^{i}$ < 60	100 ≤ RI < 200	Moderate potential risk
$60\;\leq \;E_{r}^{i}$ < 120	200 ≤ RI < 400	Considerable potential risk
$120\;\leq \;E_{r}^{i}$ < 240	400 ≤ RI < 800	High potential risk
$E_{r}^{i}$ ≥ 240	RI ≥ 800	Very high potential risk

**Table 2 ijerph-19-08380-t002:** Parameters of the human health risk assessment model.

Evaluation Parameters	Physical Meaning	Unit	Value	
			Adult	Child
*BW*	Body weight	kg	70	15
*CF*	Conversion factor	kg·mg^−1^	10^−6^	10^−6^
*IR_1_*	Ingestion rate	mg·d^−1^	100	200
*IR_2_*	Inhalation rate	mg·d^−1^	20	7.65
*ED*	Exposure duration	a	24	6
*EF*	Exposure frequency	d·a^−1^	350	350
*SA*	Skin surface area	cm	5700	2800
*AF*	Soil adherence factor	mg·cm^−1^·d^−1^	7 × 10^−2^	2 × 10^−1^
*ABS*	Dermal absorption factor	dimensionless	10^−3^	10^−3^
*PEF*	Particle emission factor	m^3^·kg^−1^	1.36 × 10^9^	1.36 × 10^9^
*AT*	Average time	d	*ED* × 365 (non-carcinogenic)	*ED* × 365 (non-carcinogenic)
			70 × 365 (carcinogenic)	70 × 365 (carcinogenic)
$ADD_{ing}$		mg·kg-day^−1^	Average daily intake by ingestion	
$ADD_{dermal}$		mg·kg-day^−1^	Average daily intake by dermal absorption	
$ADD_{inh}$		mg·kg-day^−1^	Average daily intake by inhalation	

**Table 3 ijerph-19-08380-t003:** RFD and SF values for different heavy metal exposure routes.

Heavy Metal	RFD (mg·kg^−1^·d^−1^)			SF (mg·kg^−1^·d^−1^)		
	Oral Ingestion	Dermal Absorption	Inhalation	Oral Ingestion	Dermal Absorption	Inhalation
Cu	4 × 10^−2^	1.2 × 10^−2^	4 × 10^−2^	-	-	-
Zn	0.3	0.06	0.3	-	-	-
Pb	3.5 × 10^−3^	5.25 × 10^−4^	3.52 × 10^−3^	-	8.5 × 10^−3^	-
Cd	10^−3^	10^−5^	10^−5^	6.3	-	6.3

**Table 4 ijerph-19-08380-t004:** Descriptive statistics of soil heavy metal concentrations (mg·kg^−1^).

	Cu	Zn	Pb	Cd
Background values [39]	21.90	74.00	29.10	0.08
$\bar{P}$± σ (total)	41.40 ± 1.79	74.02 ± 2.29	9.34 ± 0.35	0.35 ± 0.02
CV	0.40	0.28	0.34	0.46
Exceed multiple	1.89	1.00	0.32	4.36
national highways	38.41 ± 1.94	74.14 ± 2.46	9.71 ± 0.41	0.32 ± 0.17
CV	0.41	0.27	0.34	0.43
Exceed multiple	1.75	1.00	0.33	3.99
non-national highways	52.34 ± 3.38	73.59 ± 5.90	7.97 ± 0.58	0.46 ± 0.05
CV	0.27	0.34	0.31	0.43
Exceed multiple	2.39	0.99	0.27	5.72

**Table 5 ijerph-19-08380-t005:** Descriptive statistics of soil heavy metal concentrations (mg·kg^−1^) and their distributions near different types of roads.

Road Type	Road Number	Elements	Max (mg·kg^−1^)	Min (mg·kg^−1^)	$\bar{P}$	CV	Fold Difference
National highway	G318	Cu	82.46	14.78	43.92 ± 17.72	0.40	2.01
		Zn	123.67	53.20	85.70 ± 18.02	0.21	1.16
		Pb	14.10	5.13	8.03 ± 2.28	0.28	0.28
		Cd	0.51	0.05	0.27 ± 0.14	0.50	3.37
	G562	Cu	75.16	17.31	38.93 ± 16.97	0.44	1.78
		Zn	96.91	31.14	65.00 ± 15.77	0.24	0.88
		Pb	16.42	7.67	11.20 ± 2.57	0.23	0.38
		Cd	0.59	0.09	0.33 ± 0.13	0.39	4.09
	G219	Cu	35.65	17.81	25.71 ± 6.24	0.24	1.17
		Zn	76.97	33.44	58.00 ± 13.50	0.23	0.78
		Pb	17.99	5.49	12.42 ± 3.57	0.29	0.43
		Cd	0.62	0.23	0.39 ± 0.14	0.35	4.85
	G216	Cu	44.72	34.61	40.62 ± 3.72	0.09	1.85
		Zn	86.12	62.01	75.07 ± 7.64	0.10	1.01
		Pb	7.73	4.44	6.00 ± 1.22	0.20	0.21
		Cd	0.45	0.28	0.39 ± 0.06	0.15	4.80
Non-national highway	Gangpai Road	Cu	70.70	23.70	49.18 ± 16.69	0.34	2.25
		Zn	96.45	32.89	62.96 ± 18.24	0.29	0.85
		Pb	9.73	3.50	7.23 ± 2.55	0.35	0.25
		Cd	0.82	0.13	0.49 ± 0.24	0.48	6.10
	Laliu Road	Cu	65.02	54.78	58.65 ± 3.78	0.06	2.71
		Zn	142.12	75.48	94.84 ± 24.23	0.26	1.33
		Pb	11.85	7.73	9.45 ± 1.48	0.16	0.32
		Cd	0.47	0.32	0.40 ± 0.05	0.13	4.99

Note: $\bar{P}$ is the mean concentration of heavy metals.

## Data Availability

Not applicable.

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
