# Peer review of "Spatial Distribution of Soil Heavy Metals and Associated Environmental Risks near Major Roads in Southern Tibet, China"

_ijerph, 2022, doi:10.3390/ijerph19148380_

Round 1

Reviewer 1 Report

The subject is interesting although not very new, but my biggest criticism concerns the sampling and the interpretation of the data.

Sampling: 1) There are 14 points on 3000 km of road and highway which concerns 4 metals (CD, Cu, Zn, Pb). About the samping strategy, we do not know what is the sampling depth (1, 5, 10, 20 cm?) and if these samples really represent the soil or the deposited dust (influence of wind deposits according to the authors). One or more reference soil samples are missing (profile of one or more soils characteristics of the study area. Influence of vegetation, etc. These results are insufficient to be able to draw scientific conclusions.

2) Analysis: there is insufficient information on Atomic Absorption methods, for instance accuracy and reproducibility of measurements. For the quality of the measurements, no use of an international analytical standard is mentioned (used?). The sampling design needs to be reviewed.

3) Results: spatial distribution of metals (fig 2): for the same element, all patterns are visible (it increases or it is flat or it decreases...); This is also the caseat the same site for different metals. In view of these results, it is impossible to conclude about patterns.

4) Interpretation: The authors explain that this contamination comes from road traffic. If it is related to road traffic (which seems obvious) it would at least be necessary to have information on the traffic (number of vehicles/hour, type of vehicles (truck, car, type of fuels (gazoline, diesel). This information it is difficult to make an interpretation. Wind influence: a seasonal distribution of the different wind directions is missing to interpret the results.

Statistical indices (G-index, Single-factor potential ecological risk coefficients, Human health risk assessment, are not relevant, due to the number of samples. The discussion on the distribution of metals does not always correspond to the results (fig 2).

All of these reasons seem to me sufficient to reject this article as it stands.

Author Response

Thank you very much for the constructive comments made by the referee. According to your suggestion, we have revised the relevant issues you mentioned in the manuscript and have attached the response as an attachment.

Reviewer 2 Report

The paper is well written and the approach of assessment of environmental risks is correct. The information contained in this work is valuable and interesting from the perspective of environmental hazards due to soil pollution with heavy metals related to traffic sources.

The topic is very important for many categories of readers and the paper can be accepted after the following corrections and suggestions are carefully met:

1) In several lines (105,120,160,176,187, 194-195, 209,224, 236-237,248-249, 253, 262, 281,286,324-325) it should be removed/corrected: “Error! Reference source not found.

2) At lines 391-392, to rephrase: “And the combined potential ecological risk level was considerable.

3) All references should be cited according to the journal requirements.

Author Response

(The authors gave the same response as above.)

Reviewer 3 Report

Manuscript ID: Int. J. Environ. Res. Public Health 2022, 19, x. https://doi.org/10.3390/xxxxx

Title: Spatial distribution of soil heavy metals and the associated environmental risks near major roads of southern Tibet, China

The manuscript concerns the contamination of soil heavy metals (Cu, Zn, Pb, Cd) near six roads in the southern part of the Tibetan Plateau.

In general, the manuscript has correct structure, is clearly written and based on current literature.

Material and methods - missing basic data on the laboratory analysis, validation (details below).

Many times in the text - Error! Reference source not found – I cannot verify this.

Abstract section:

Correct.

Keywords:

Correct.

Introduction section:

Correct.

Materials and methods section:

L124-126 – I suggest to add some  description of the method of the sample digestion (chemicals, methods, conditions, procedures). Wat is HJ491-2019 and GB/T17141-1997 – I can’t find in References. What are the QA/QC steps taken throughout the analysis? Any standards, reference materials? Is it the total content of the investigated elements or some kind of fraction – mobile?

Rest is correct.

Results section:

L194-195 – where is it listed in - Error! Reference source not found.??;

L195 – where is the “table” 4?;

L 197 – where these values are shown ? 3.99-, 1.75-, and 1.00-fold

L199 – and here? 5.72-, 2.39-, and 0.99-fold the background values – what is the background?

Too many “Error! Reference source not found” – I cannot verify this.

Discussion section:

Correct

Conclusions section:

Correct.

References section:

Correct.

Tables and figures section:

Correct.

Finally:

In my opinion the article is very interesting and comprehensive. I recommend that the manuscript be accepted after minor revision.

Author Response

(The authors gave the same response as above.)
